# Cell cycle population effects in perturbation studies

Eoghan O'Duibhir[†], Philip Lijnzaad[†], Joris J Benschop, Tineke L Lenstra, Dik van Leenen, Marian JA Groot Koerkamp, Thanasis Margaritis, Mariel O Brok, Patrick Kemmeren & Frank CP Holstege[*]

## Abstract

Growth condition perturbation or gene function disruption are commonly used strategies to study cellular systems. Although it is widely appreciated that such experiments may involve indirect effects, these frequently remain uncharacterized. Here, analysis of functionally unrelated *Saccharyomyces cerevisiae* deletion strains reveals a common gene expression signature. One property shared by these strains is slower growth, with increased presence of the signature in more slowly growing strains. The slow growth signature is highly similar to the environmental stress response (ESR), an expression response common to diverse environmental perturbations. Both environmental and genetic perturbations result in growth rate changes. These are accompanied by a change in the distribution of cells over different cell cycle phases. Rather than representing a direct expression response in single cells, both the slow growth signature and ESR mainly reflect a redistribution of cells over different cell cycle phases, primarily characterized by an increase in the G1 population. The findings have implications for any study of perturbation that is accompanied by growth rate changes. Strategies to counter these effects are presented and discussed.

**Keywords** environmental stress response; gene deletion; gene expression; genome-wide transcription; growth rate
**Subject Categories** Genome-Scale & Integrative Biology; Cell cycle; Methods & Resources
**Mol Syst Biol. (2014) 10: 732**

## Introduction

Perturbation is a universally applied approach to study the behavior and molecular mechanisms underlying cellular systems (Ideker *et al*, 2001). A perturbation can be environmental, for example a change in growth condition or the addition of a response-inducing compound. A perturbation can also be the targeted disruption of a particular cellular component, for example by gene deletion or through RNA-mediated knockdown. Combinations of these two general types of perturbation are also frequently applied. The range of possible readouts that can be monitored to study systems properties is extremely large. Dependent on the system being studied and the questions being asked, readouts can vary between a relatively simple phenotype such as growth, to the expression levels of all genes. Given the interconnected nature of cellular systems (Bensimon *et al*, 2012; Walhout & Vidal, 2001), any systems readout is potentially determined by a combination of different types of direct and indirect mechanisms. Examples include ascribing a role as direct regulator to a particular transcription factor that is in fact high up in a regulatory cascade of other transcription factors more directly responsible for the readout (Spitz & Furlong, 2012). For environmental perturbations, an altered nutrient environment may have an indirect influence on a phenotype such as cell size due to a more direct effect on doubling time for example (Zaman *et al*, 2008). There are therefore many different types of indirect effects. Depending on the goals of a particular study, such effects need to be taken into account, especially if the goal is to derive molecular mechanisms.

Here, we further analyze a dataset describing changes in genome-wide expression patterns for 1,484 *Saccharomyces cerevisiae* gene deletion strains (Kemmeren *et al*, 2014). We describe a gene expression signature common to many of the strains, with subsequent analyses aimed at determining the cause of the common expression signature. We show that the signature is similar to the environmental stress response (ESR) gene expression signature, previously described as a cellular response exhibited upon many different environmental perturbations such as nutrient limitations and different types of stress (Brauer *et al*, 2008; Gasch *et al*, 2000). Further analyses show that both the ESR and the expression signature common to slow growing deletion strains result to a large extent from shifts in the proportions of cells in different cell cycle phases. The results have implications for any study applying environmental or genetic perturbations that result in growth rate changes.

## Results

### A recurrent gene expression signature exhibited by slow growing deletion mutants

We have previously carried out whole-genome mRNA expression profiling of 1,484 *S. cerevisiae* single gene deletion strains grown

---

Molecular Cancer Research, University Medical Center Utrecht, Utrecht, the Netherlands
*Corresponding author. Tel: +31 88 755 5874; Fax: +31 88 756 8531; E-mail: f.c.p.holstege@umcutrecht.nl
†These authors contributed equally to this work

under identical conditions (Kemmeren *et al*, 2014). Deletion strains were selected based on a (putative) role as regulator of gene expression, also using characteristics such as nuclear location or the capacity to modify other proteins. This results in coverage of diverse categories such as gene-specific and global transcription factors, RNA processing and export, ubiquitin-like modifications, protein kinases/phosphatases, protein trafficking, cell cycle, meiosis, DNA replication and repair. Of the 1,484 deletion strains, 700 exhibit an expression profile that is robustly different from wild-type: more than three transcripts with expression changed over 1.7-fold and with $P < 0.05$ compared to the average of over 400 wild-type strains, (Kemmeren *et al*, 2014). Analysis of the mutants with robustly altered transcriptomes shows that alongside specific expression changes, many strains display a shared expression signature (Fig 1A). In order to study this recurrent signature, it was separated from specific effects using principal component analysis (PCA, Materials and Methods). Projection of the original gene expression data (Fig 1A) onto the first principal component axis demonstrates the recurrent nature of this signature (Fig 1B, Supplementary Dataset S1). The first principal component is distinct from other components as it accounts for 24% of the variation in the entire dataset (Supplementary Fig S1) and is shared by 25% ($r > 0.5$) of the 700 robustly affected deletion mutants.

Genome-wide expression changes that are shared between two or more mutants typically indicate similar function such as shared protein complex or pathway membership (Benschop *et al*, 2010; Hughes *et al*, 2000; Kemmeren *et al*, 2014; Lenstra *et al*, 2011; Roberts *et al*, 2000). In contrast, this common signature is found as part of the expression profile of many different mutants that also belong to diverse functional groups (Fig 1C). What is shared by the majority of these strains is a reduction in growth rate, with more slowly growing strains exhibiting a stronger presence of the recurrent signature (Fig 1D, Supplementary Dataset S2). In other words, the doubling time of the various deletion strains correlates with the degree to which the recurrent signature is present. This finding agrees with the previously reported observation that the number of genes with changed expression scales with the degree of slower growth (Brauer *et al*, 2008; Hughes *et al*, 2000; Keren *et al*, 2013; Regenberg *et al*, 2006). It is also important to note that the apparent poor correlation of some deletion mutants to the recurrent signature (Fig 1D, off-diagonal points) is due to additional gene expression changes specific to particular mutants (examples in Supplementary Fig S2). While not identifying which aspect is causative, the correlation between growth rate and the presence of

the recurrent signature indicates a link between growth rate in deletion mutants and expression changes in those genes most strongly affected.

**The recurrent signature is not caused by medium depletion**

All strains were grown for two cell doublings prior to harvesting (Fig 2A, left). This is sufficient for recovery from overnight preculture and also allows slow growing strains to achieve balanced, exponential growth (Supplementary Fig S3). An initial concern was that the environment of slower growing strains at the time of harvest may have been different compared to wild-type because of a longer time spent in culture. The medium of a slower growing strain may have been more depleted of nutrients. Alternatively, compounds may have been excreted into the medium for a longer time by the slower growing strains. Such growth rate-dependent changes to the media may account for the slow growth gene expression changes, in particular because of the correlation between the magnitude of the signature and the growth rate reduction (Fig 1D). To investigate this possibility, we expression-profiled wild-type strains grown for two cell doublings in media pre-conditioned by culture of slow growing deletion strains (Fig 2A). The expression profiles of the deletion strains are shown in Fig 2B–E. Wild-type yeast grown in the pre-conditioned media from the deletion strains do not display the gene expression changes observed for the deletion strains grown previously in the same medium (Fig 2F–I). Growth rate-dependent changes to the media therefore do not account for the slow growth associated signature.

**The slow growth signature is pervasively present in genome-wide expression studies**

To better understand the nature of the slow growth associated gene expression signature, we next systematically searched publicly available yeast gene expression datasets for correlations with the recurrent profile. The slow growth signature was highly correlated with many previously published microarray datasets (Fig 3A). Curiously, rather than being restricted to studies of deletion strains, correlations were also found with studies of wild-type cells cultured under different growth conditions. Several of the strongest correlations are with wild-type cells grown under various conditions (Causton *et al*, 2001; Gasch *et al*, 2000) (Fig 3A). In the Gasch *et al* study, a large gene expression signature was described that

---

**Figure 1. Recurrent signature in deletion strains associated with reduced growth rates.**

A   Heat map of mRNA expression changes in the 700 deletion mutants that display the most robust changes in gene expression [more than three transcripts changing more than 1.7-fold and with *P*-value < 0.05 compared to the average of over 400 wild-types (Kemmeren *et al*, 2014)]. Both the transcripts and deletion strains have been clustered (cosine correlation; average linkage).

B   The recurrent signature of the dataset shown in (A), plotted as the projection of the first principal component and its presence in each deletion strain profile. Transcripts and deletions strains are ordered as in (A).

C   The occurrence of the recurrent signature according to functional category of the deleted gene in each strain. Shown in blue are the fractions when considering only strains displaying a strong recurrent signature, here defined as deletion strains having a correlation greater than 0.5 with the signature. Only 'protein trafficking' ($P = 4.5 \times 10^{-3}$) and 'cell cycle regulation' ($P = 0.018$) are significantly ($P < 0.05$) overrepresented among strains with a strong recurrent signature (hypergeometric test, Bonferroni-corrected).

D   The similarities of deletion profiles to the recurrent signature, versus their growth rate plotted as $\log_2$ (doubling time in mutant/doubling time wild-type). The similarity is expressed as the projection of a deletion profile onto the normalized recurrent profile (this is proportional to their covariance). The blue dots show the deletion mutants further analyzed by flow cytometry in Fig 5.

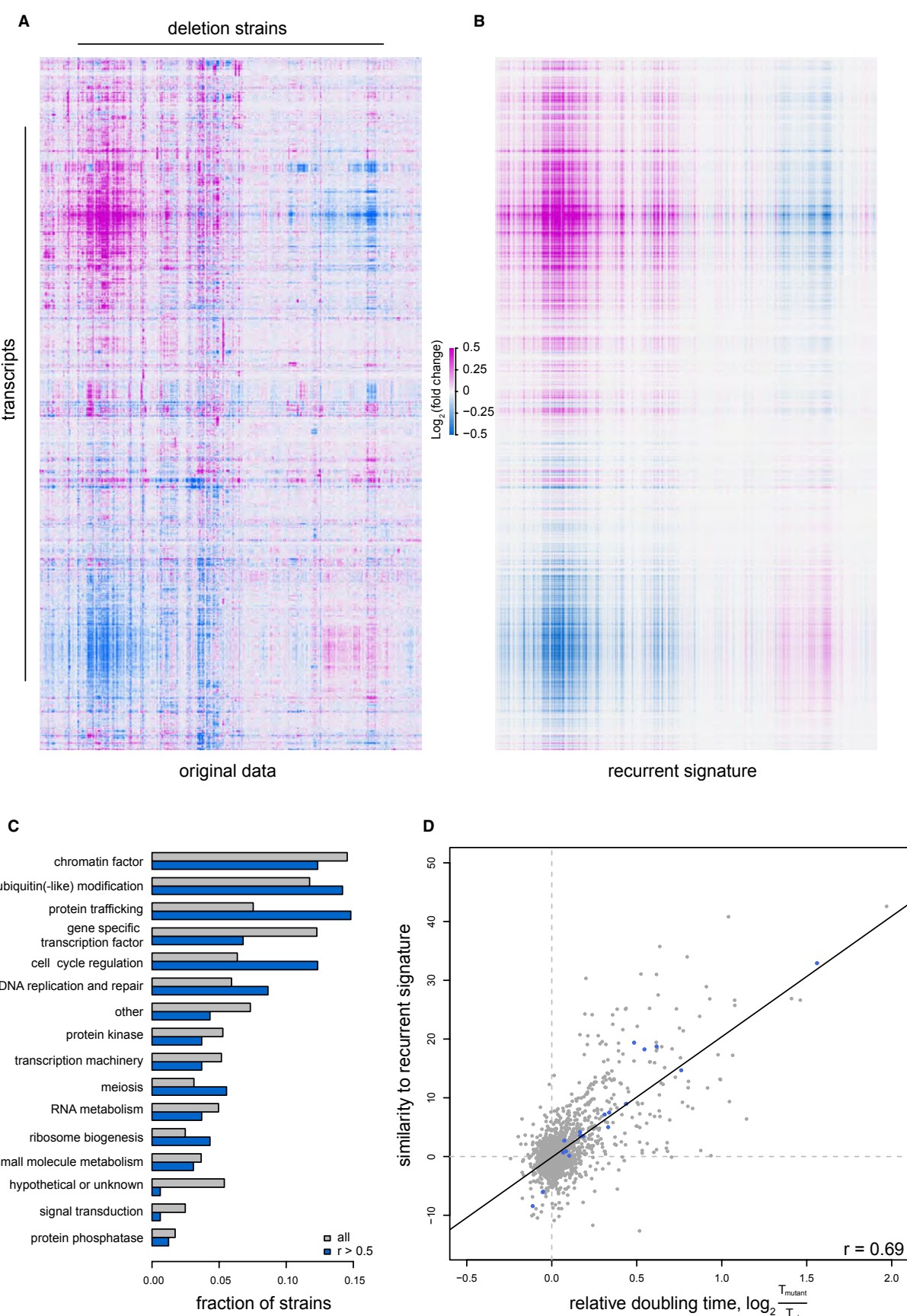

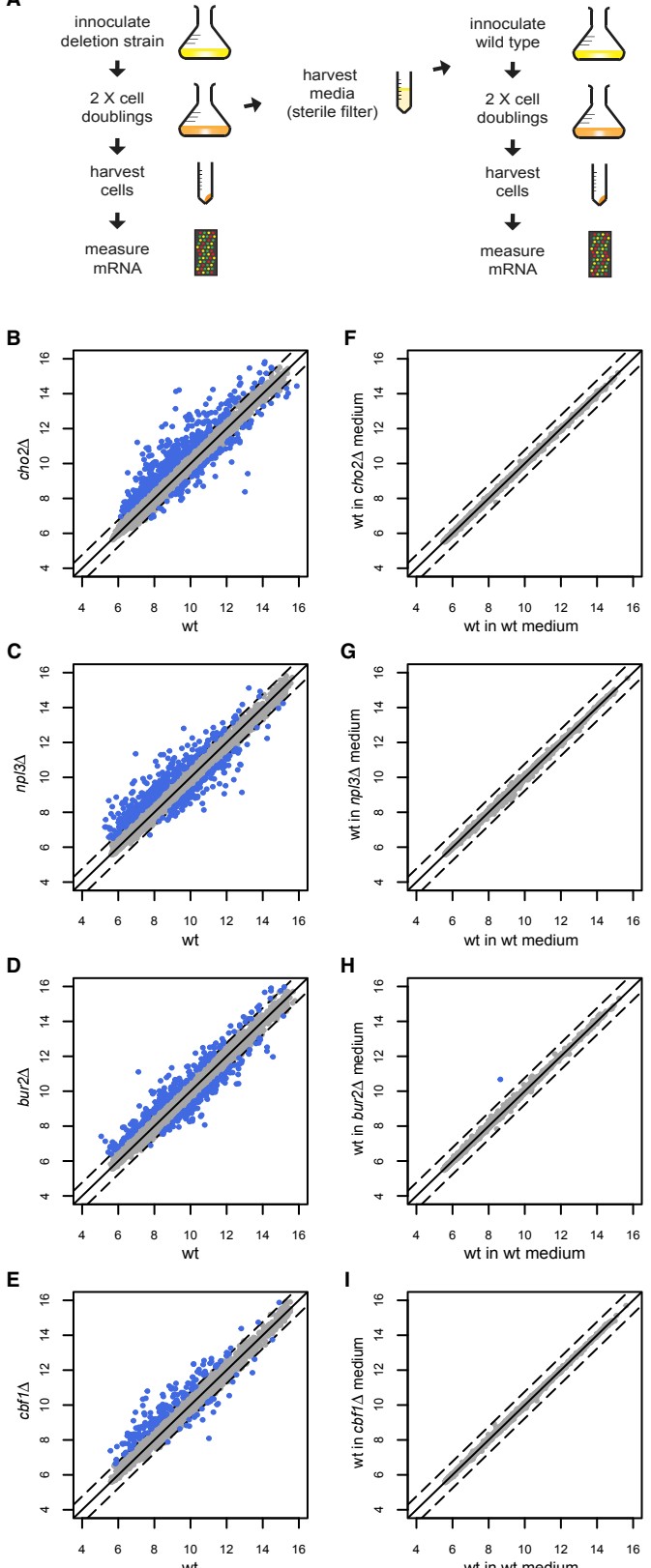

occurs under all conditions of stress tested: the environmental stress response (ESR). Of all the stress conditions examined, heat shock (15 min after shift from 29°C to 37°C (Gasch *et al*, 2000)) gave the highest correlation to the slow growth signature (Fig 3B, *r* = 0.73). This correlation increases further upon repeating the heat shock experiment with our own platform and genetic background (Fig 3C, *r* = 0.82). The ESR was not defined on heat shock alone. Rather, the ESR is characterized by genes that change in expression during many environmental perturbations including addition of hydrogen peroxide, menadione, diamide, DTT, osmotic shock and various nutrient limitations (Gasch *et al*, 2000). When only considering those genes that define the ESR, the correlation with the slow growth signature increases even further (Fig 3D, *r* = 0.93). High correlations are also found under different conditions analyzed with different technologies (Fig 3E and F). These analyses show that the signature that is common to slower growing deletion strains is highly related to the ESR signature shown by wild-type cells subjected to many different types of growth condition perturbation.

### The ESR signature can be explained by a cell cycle population shift

To determine whether there is a mechanism common to the environmental and genetic perturbations that result in a shared expression signature, wild-type response to environmental perturbation was first studied in more detail. Many stressful conditions, including heat shock, osmotic stress, and DNA damage, cause a transient G1 arrest (Bellí *et al*, 2001; Gerald *et al*, 2002; Johnston & Singer, 1980; Rowley *et al*, 1993). Gene expression measurements typically describe the average expression levels of genes in the population of cells that make up the culture. In unsynchronized exponentially growing batch cultures, this population is made up of cells rapidly progressing through the cell cycle. The measured gene expression level is therefore the average across the entire cell cycle. Cells subjected to mild heat shock transiently arrest their progress through the cell cycle at the START checkpoint, between early and late G1 (Johnston & Singer, 1980; Rowley *et al*, 1993). A transient G1 arrest due to heat shock would change the distribution of cells over each of the cell cycle phases. To determine whether a shift in the distribution of cells over different cell cycle phases can explain the ESR, an *in silico* approach was taken. Cell cycle gene expression time-course data from cells synchronized by elutriation (Spellman *et al*, 1998) were summed using weights that represent the fraction of cells in each respective cell cycle phase, yielding a virtual profile that simulates a population of unsynchronized cells. The weights

---

**Figure 2. The slow growth associated signature is not caused by an altered culture medium in slower growing strains.**

A    Overview of the experimental procedure to determine any influence of the media.

B–I    Scatter plot expression profiles of deletion strains versus wild-type (left column) and those of wild-type strains (right column) grown in media in which the corresponding deletion strain was first grown. The numbers on the axis refer to the averaged log$_2$ fluorescent dye intensities of the microarray probes representing each gene (dots). The dashed line indicates a 1.7-fold-change in mutant versus wild-type. Genes with *P* < 0.05 and fold-change over 1.7 versus wild-type are colored blue.

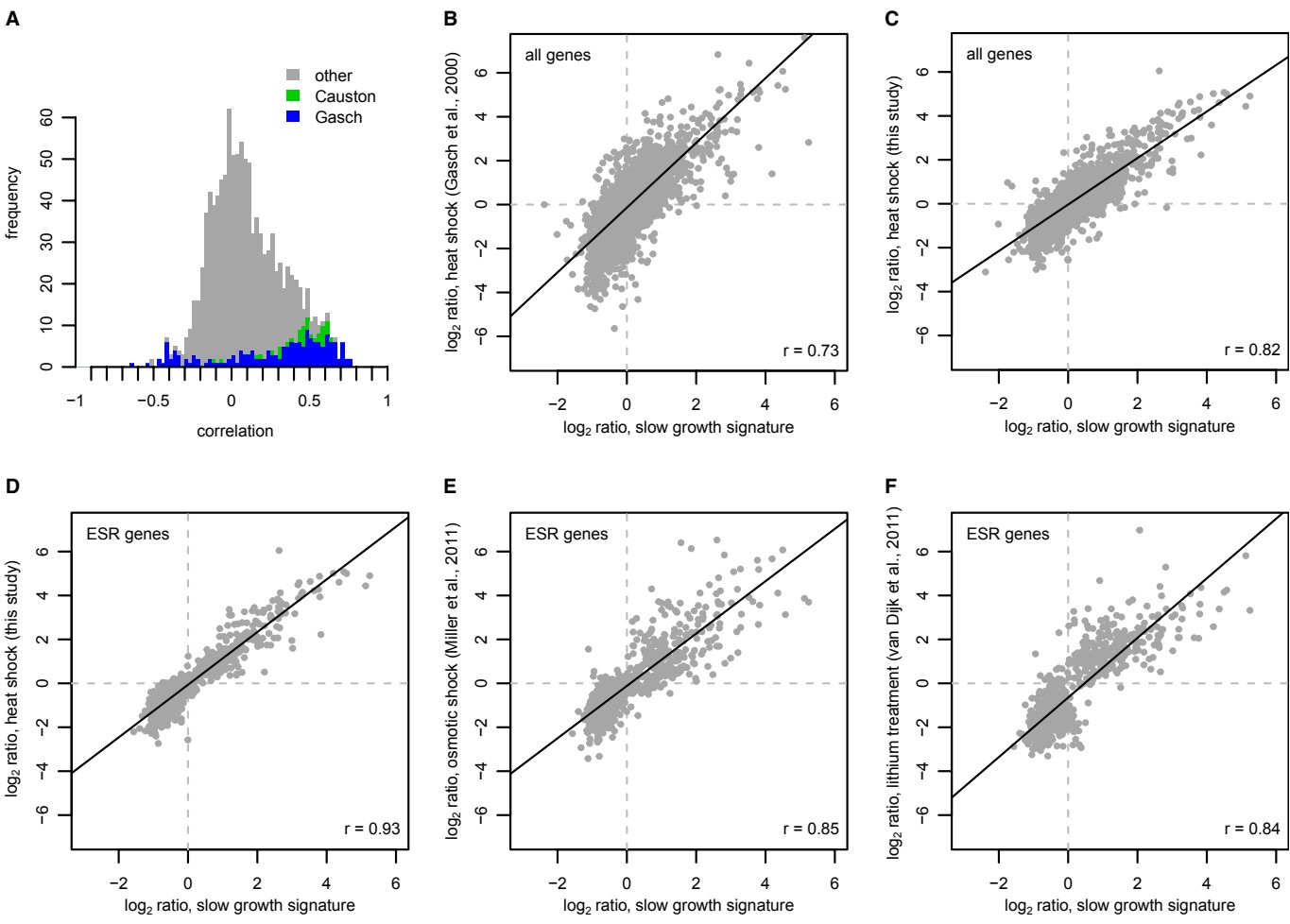

**Figure 3.  The slow growth associated signature is similar to the environmental stress response (ESR).**

A    Distribution of correlations with the recurrent signature for a compendium of expression profiles from literature (Backhus *et al*, 2001; Barbara *et al*, 2007; Bernstein *et al*, 2000; Causton *et al*, 2001; Chu *et al*, 1998; Dasgupta *et al*, 2002, 2004; DeRisi *et al*, 1997; Epstein *et al*, 2001; Fazzio *et al*, 2001; Ferea *et al*, 1999; Galitski *et al*, 1999; Gasch *et al*, 2000; Hardwick *et al*, 1999; Holstege *et al*, 1998; Hu *et al*, 2007; Hughes *et al*, 2000; Jelinsky *et al*, 2000; Lee *et al*, 2000; Madhani *et al*, 1999; Ostapenko & Solomon, 2003; Primig *et al*, 2000; Roberts *et al*, 2000; Roth *et al*, 1998; Spellman *et al*, 1998; Travers *et al*, 2000; Vary *et al*, 2003; Viladevall *et al*, 2004; Young *et al*, 2003; Zhu *et al*, 2000). Indicated in blue are the correlations against the profiles described in Gasch *et al* (2000) and in green, those by Causton *et al* (2001).

B–F    Scatter plots comparing the slow growth signature to: (B) the (Gasch *et al*, 2000) heat shock expression profile at *t* = 15 min; (C) a heat shock expression profile at *t* = 15 min performed with an identical strain background, growth medium, microarrays and other procedures as in Fig 1; (D) heat shock (from C) when considering only ESR genes, that is, those defined previously to form the environmental stress response (Gasch *et al*, 2000); (E) the expression changes of ESR genes 36 min after osmotic shock as determined using Affymetrix short oligonucleotide arrays (Miller *et al*, 2011); and (F) expression changes of ESR genes after 100 mM lithium addition as determined using RNA-seq (Van Dijk *et al*, 2011).

were determined by a single spline, controlled by four parameters which were optimized for correlation to the (Gasch *et al*, 2000) heat shock gene expression profile (15 min, 29 to 37°C). Strikingly, when weighted in this manner, the cell cycle gene expression data closely resemble the ESR ($r$ = 0.88, Fig 4A).

To rule out overfitting, the same procedure was performed on randomized ESR genes or on randomized cell cycle expression data, both yielding no correlation upon optimization (average $r$ = 0.06 and 0.05 respectively, Materials and Methods). Furthermore, multiple cross-validations with half of the ESR genes used for fitting the cell cycle data, and applying this fit to the remaining ESR genes yields average correlations of 0.88 for the heat shock (Materials and Methods). Cell cycle synchronization can be

achieved in a variety of ways. As discussed by Shedden and Cooper (Shedden & Cooper, 2002), the elutriation dataset is the least prone to stress and all methods suffer from quite rapid loss of synchronization. When other cell cycle time series are used, the results are nevertheless similar. For elutriation from Spellman *et al* (1998), the correlation between heat shock and the cell cycle based model (Fig 4A) is 0.88. For alpha factor arrest from Spellman *et al* (1998) and Granovskaia *et al* (2010), the correlation is 0.82 and 0.63 respectively. For *cdc28-ts* (temperature-sensitive allele) from Spellman *et al* (1998) and Granovskaia *et al* (2010), the correlation is 0.71 and 0.35 respectively. The ability to mimic the ESR from cell cycle time-course data is a strong indication that a large part of the ESR derives from a cell cycle population shift

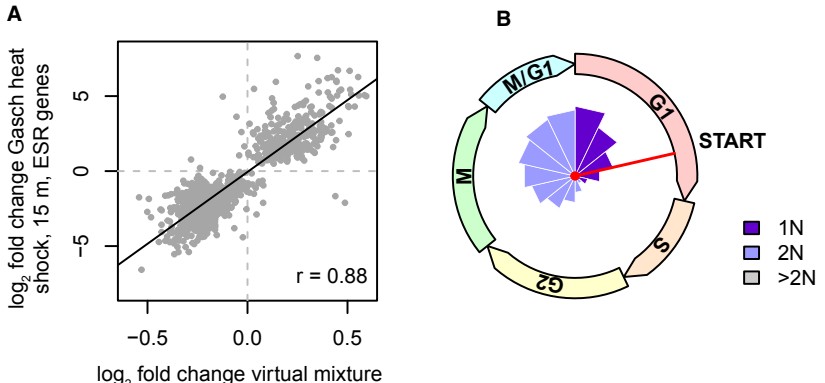

**Figure 4.  Cell cycle population shift underlies the ESR.**

A    Scatter plot comparing the expression profile of *t* = 15 min heat shock (Gasch *et al*, 2000) with the spline-weighted mix of cell cycle phase signatures (Spellman *et al*, 1998). Absence of data points around 0 is due to use of ESR genes, which by definition are those with changes.

B    Circular histogram of the cycle phase population distribution used in (A). The red line shows the START checkpoint. Names of cell cycle phases are shown on the dial.

C    Snapshots of the simulated cell cycle phase population distribution upon heat shock.

D    Time series of flow cytometry profiles after heat shock.

E    Modeled (line) and observed (dots) fraction of 1N cells during heat shock.

F    Heat map of mRNA expression changes during a heat shock time series including all transcripts changing more than 1.7-fold and with *P* < 0.05 in any single time point compared to *t* = 0. Scale as in Fig 1.

G    Modeled (line) and observed (dots) average magnitude of gene expression changes for genes with increased (purple) or decreased (blue) expression during heat shock.

rather than from a direct cellular transcriptional response to environmental perturbation.

The cell cycle distribution derived from weighting of the cell cycle gene expression data predicts that the ESR involves an increase in the number of cells in early G1 and a decrease in the number of cells in late G1 and S phase (Fig 4B). Importantly, this fits with the transient arrest at START encountered upon mild heat shock (Johnston & Singer, 1980; Rowley *et al*, 1993). To test this further, a predictive model was created, whereby a transient arrest of cell cycle progress was initiated at *t* = 0. This corresponds to the heat shock induced cell cycle arrest at START (Rowley *et al*, 1993). In the model, this results in accumulation of early G1 cells and concomitant depletion of late G1 cells (Fig 4C and Supplementary Movie S1). Because both early and late G1 cells have 1N DNA content, the model predicts that the fraction of 1N cells will begin to accumulate later during heat shock and not immediately (Fig 4E, line). To test the model, flow cytometry analysis of a heat shock time-course experiment was performed. The result agrees with previous analyses (Johnston & Singer, 1980; Rowley *et al*, 1993) and exhibits a 1N accumulation both qualitatively and quantitatively close to the values predicted by the model (Fig 4E, dots; individual flow cytometry profiles, Fig 4D). In agreement with the finding that the ESR can be modeled by weighting of cell cycle gene expression data (Fig 4A), the cytometry analysis shows that a cell cycle population shift indeed takes place, also in the form predicted.

As with the shift in cell cycle populations modeled and measured during heat shock (Fig 4C–E), the ESR gene expression signature is also transient (Gasch *et al*, 2000). As a further test of the idea that the ESR signature is to a large extent caused by a shift in the distribution of cells over different cell cycle phases, the population model (Fig 4C) was used to predict the average magnitude of gene expression changes. This model simply assumes that the magnitude of all gene expression changes is directly proportional to the change in

distribution of cells in each cell cycle phase as compared to the unperturbed steady state distribution at *t* = 0. A transient arrest of 25 min was incorporated as this is the timeframe previously shown to be required for maximal expression of chaperone proteins (Miller *et al*, 1982), allowing adaptation to the higher temperature. Actual gene expression changes were then monitored during the course of a heat shock experiment (Fig 4F). Despite the simplicity of the model, the predicted transient changes (Fig 4G, lines) globally fit the average magnitude changes measured (Fig 4G, dots). The ability to recreate the ESR *in silico* from cell cycle phase signatures of unstressed cells (Fig 4A), the agreement between predicted and observed cell cycle population shifts that actually occur (Fig 4C–E) and the predicted and observed transient nature (Fig 4G) all support the idea that the ESR signature to a large extent reflects a cell cycle population shift.

### A cell cycle population shift also underlies the recurrent slow growth signature observed upon genetic perturbation

Growth rate change accompanies many environmental perturbations (Bellí *et al*, 2001; Gerald *et al*, 2002; Johnston & Singer, 1980; Rowley *et al*, 1993). Growth rate reduction also accompanies many single gene deletions (Fig 1D, Supplementary Dataset S2). An expression signature similar to the ESR has been observed in wild-type continuous culture experiments forced to grow slowly (Brauer *et al*, 2008; Regenberg *et al*, 2006). The recurrent slow growth signature found in many deletion strains is also highly similar to the ESR (Fig 3). We therefore reasoned that the similar signatures observed in both cases could be the result of the same phenomenon: a shift in the cell cycle distribution of the population. To test this possibility, deletion strains covering a wide range of growth rates (blue dots, Fig 1D) were analyzed by flow cytometry (Fig 5A and B). The degree of presence of the slow growth expression signature is

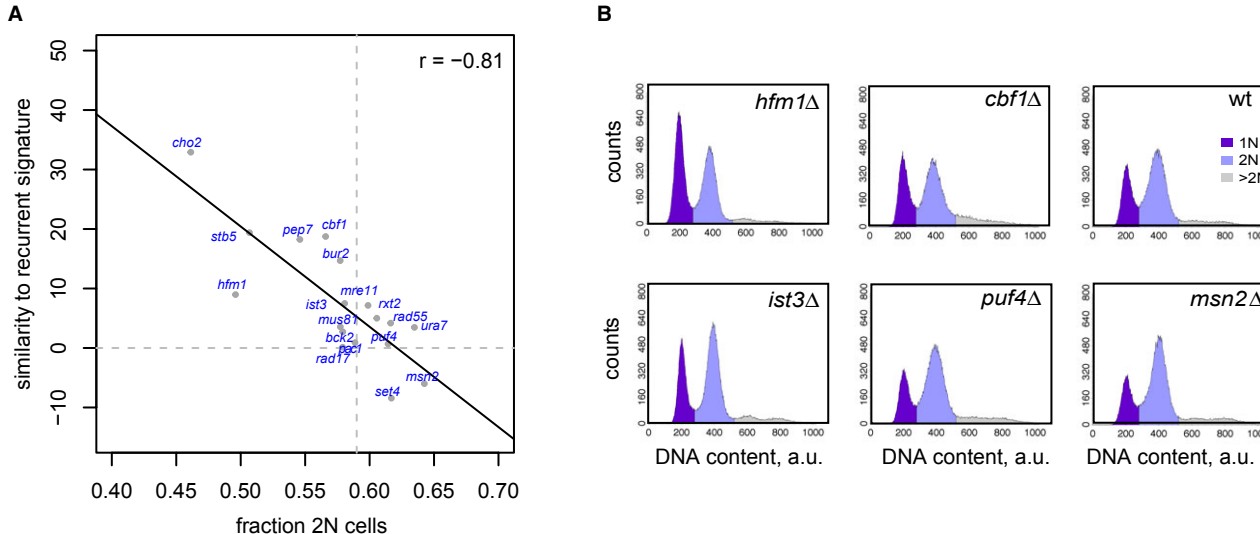

**Figure 5.   Cell cycle population shifts and their correlation with the slow growth signature in deletion strains.**

A   Scatter plot comparing the similarities of gene deletion expression profiles to the slow growth signature, versus the fraction of 2N cells as determined by flow cytometry. Dashed lines indicate the wild-type values.

B   Examples of flow cytometry profiles used in (A).

clearly proportional to the degree of cell cycle population shift observed for the individual strains (Fig 5A). As with the ESR signature, the common signature found in slower growing deletion strains also therefore results to a large extent from a shift in the cell cycle distribution.

### Identifying primary effects of genetic and environmental perturbation

For those perturbations that result in growth rate reduction, it is likely that the reduction in growth rate will often be a more downstream consequence. In such cases, removal of the gene expression signature associated with slower growth may help reveal the more immediate and direct effects of either an environmental or genetic perturbation. To test this, we transformed a previously published gene expression dataset from cells grown under amino acid starvation (Gasch *et al*, 2000), by factoring out the slow growth signature (Materials and Methods). Transformation of the amino acid starvation data in this way (Fig 6A) results in a more pronounced enrichment for genes involved in amino acid biosynthesis compared to the

original data. The same approach was also applied to an environmental perturbation response not included in the Gasch study: growth in low phosphate (Fig 6B). Here, transformation of the data enriches for finding direct targets of the low phosphate activated transcription factor Pho4. Transformation of environmental perturbation datasets in this way can therefore improve delineation of direct responses.

The approach was also tested for genetic perturbation. For this, we analyzed gene-specific transcription factor (GSTFs) deletion profiles in the starting dataset (Fig 1) for which there was also genome-wide binding data available (MacIsaac *et al*, 2006). Similar to environmental perturbation, transformation of the GSTF deletion data by removal of the slow growth associated signature improves identification of direct target genes. This is evident from the reduced false positive rate for finding GSTF binding in the promoters of genes with changed expression upon GSTF deletion (Fig 6C). Importantly, the improvement is greatest for the subset of GSTF deletions that suffer most from slow growth and does not interfere adversely for deletions that are less affected by slow growth (Fig 6C and Supplementary Fig S4). Transformation of

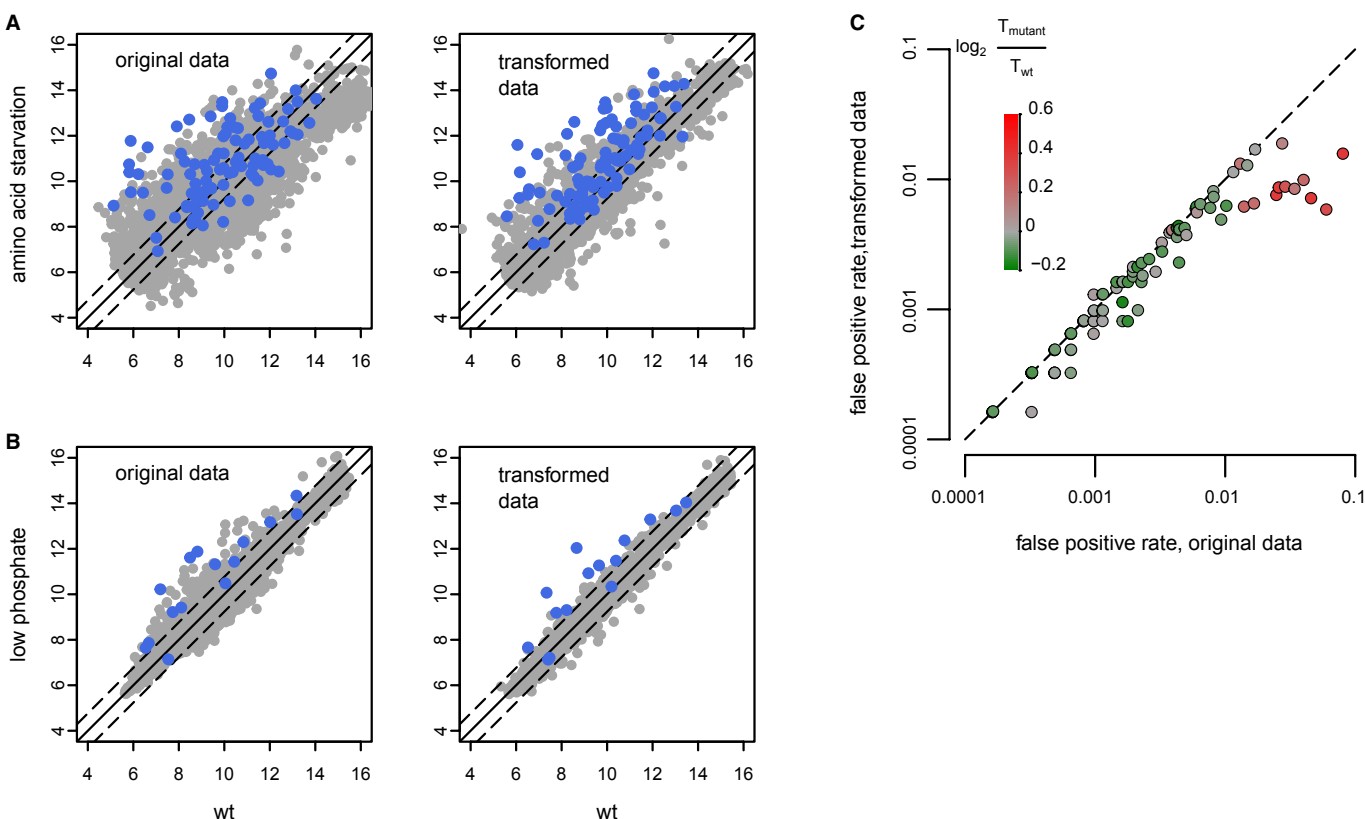

**Figure 6.  Transformation of expression data to identify more direct consequences of perturbation.**

A  Expression changes under amino acid starvation conditions (30 min, Gasch *et al*, 2000), before (left) and after (right) transformation (removal of the first principal component). Transcripts of genes annotated with "cellular amino acid biosynthetic process" (GO:0008652) are in blue ($P = 5.7 \times 10^{-10}$ original, $P = 2.9 \times 10^{-24}$ transformed: using a hypergeometric test for enrichment among genes increasing more than 1.7-fold in expression with $P < 0.05$).

B  Expression changes in wild-type strain at low (75 μM) versus standard (10 mM) phosphate conditions, before (left) and after (right) transformation. Targets of the low phosphate transcription factor Pho4 (Ogawa *et al*, 2000) are in blue ($P = 4.5 \times 10^{-16}$ original, $P = 2.3 \times 10^{-24}$ transformed).

C  False positive rate comparison before and after transformation of transcription factor binding site enrichment analyses for 100 gene-specific transcription factor deletion strains. Strains are colored with respect to their $\log_2$ relative doubling time as shown by the scale.

gene expression datasets can therefore improve delineation of direct effects for both environmental as well as genetic perturbations. The fact that transformation works for both types of perturbations also supports the idea of a common underlying secondary response and fits the proposal that cell cycle population shifts should be taken into account in any study of perturbations that result in growth rate changes.

## Discussion

The finding of a slow growth associated expression signature in deletion strains fits with previous reports linking global changes in gene expression to changes in growth rates, in particular growth rate changes caused by various nutrient limitations (Brauer *et al*, 2008; Keren *et al*, 2013; Regenberg *et al*, 2006). Here, we describe an expression signature common to slow growing deletion strains. As with wild-type strains growing more slowly due to particular growth conditions, the signature common to slow growing deletion strains is highly similar to the ESR, despite the deletion strains being grown under optimal conditions. Besides a shared expression response, the feature that is common to all these different perturbations (genetic, nutrient and the various forms of stress) is growth rate reduction. Many stressful conditions, including heat shock, osmotic stress, and DNA damage, cause a transient G1 arrest (Bellí *et al*, 2001; Gerald *et al*, 2002; Johnston & Singer, 1980; Rowley *et al*, 1993). Indeed, detailed examination of the expression data used to originally define the ESR reveals a transient reduction in the mRNA levels of the post START pre-S phase marker *CLN2* in the vast majority of the stress time courses studied (Supplementary Fig S5). Disruption of cell cycle progression has previously been indicated to cause many indirect gene expression changes (Lu *et al*, 2003). The latter study also indicated a link between cell cycle population effects and gene expression, thereby providing the basis for the ESR/cell cycle modeling approach taken here. The modeling and flow cytometry analyses (Figs 4 and 5) show that perturbations resulting in slow growth have an altered cell cycle population distribution and that this explains a large part of the commonly found expression response.

An alternative explanation is that all the different perturbations result in stress of which the common expression signature (slow growth/ESR) is part and that this causes the reduced growth rate. This does not fit with the ability to model the ESR using cell cycle phase signatures (Fig 4). Furthermore, the recent demonstration that commitment to the mitotic cell cycle occurs before induction of G1-S gene expression changes (Eser *et al*, 2011), also rules this alternative out. Rather than constituting a genome-wide response in single cells, the ESR and the deletion strain slow growth signature are therefore mainly a manifestation of a response at the level of the culture as a whole.

The correlations achieved by modeling a population shift are high but not complete (Fig 4). This indicates that while the majority of the ESR can be explained by the cell cycle population shift, there is also an independent component. This likely corresponds to the well-documented transcriptional response to general stress mediated by transcription factors such as Msn2/4 (Morano *et al*, 2012). Direct targets of Msn2 are indeed part of both the ESR (Gasch *et al*, 2000) and the slow growth signature (Supplementary Table S1). The

presence of distinct contributions to the ESR has been suggested before (López-Maury *et al*, 2008). Further work to completely unravel the contributions of the distinct components will potentially have to take into account the possibility of metabolic/redox cycles also causing cyclic expression of general stress response target genes (Chen *et al*, 2007; Tu *et al*, 2005). Despite these complicating factors, the part of the ESR previously linked to growth (Brauer *et al*, 2008; López-Maury *et al*, 2008) can now be attributed to the cell cycle population shift.

A practical outcome of this study is that the extensive scope of the genetic perturbation data provides a means to calculate a high quality vector representation of the recurrent slow growth signature (Supplementary Dataset S4). The vector can be used to determine which transcripts are most strongly influenced by slow growth. Gene ontology (GO) analysis of the signature (Supplementary Table S1) shows enrichment for many of the same processes previously described in detail for the ESR (Gasch *et al*, 2000). The enriched categories include many metabolic processes intimately coupled to cell cycle progression (Chen *et al*, 2007; Tu *et al*, 2005), in agreement with a cell cycle population shift underlying a large part of these signatures.

Subtraction of the slow growth vector from the gene expression patterns of genetically or environmentally perturbed cells improves identification of more direct responses to the perturbation (Fig 6). The signature is present in datasets generated with diverse platforms (Fig 3). Although transformation of data across different technology platforms is possible (Fig 6A), this likely works better within a single platform (Irizarry *et al*, 2005). A transformed dataset for all gene deletion expression profiles is made available (Supplementary Dataset S5), as is a file combining the original and transformed data including the *P*-values relating to the original measurements (Supplementary Dataset S6). Weighting of the degree of data transformation differs depending on the degree to which the slow growth signature is present. This also fits with the observation that the degree of reduced growth differs for each deletion strain and for different environmental perturbations. In agreement with the idea that the recurrent signature is a result of cell cycle population shifts, it is interesting to note that the inverse signature is also present in a subset of deletion strains (Fig 1B and D) and that these also show the opposite cell cycle population shift (Fig 5A, *msn2Δ* and *set4Δ*). The same holds for a subset of environmental perturbation data (Fig 3A, negative correlations). One of these is a cold shock transfer from an adverse 37°C to a more optimal growth temperature of 25°C (Gasch *et al*, 2000). Cold shifted cells also show the opposite flow cytometry profile shift (Supplementary Fig S6), all in line with the explanation that the recurrent signature is a reflection of altered cell cycle phase populations.

It is unlikely that all analyses will benefit from data transformation to remove the recurrent signature. The presence of slow growth, a cell cycle population shift, and its associated signature are phenotypes. Just like other phenotypes, for analyses of similarity between different deletion strains, the presence of such characteristics can help place mutants into similarly behaving groups. Data transformation should therefore only be applied in studies requiring identification of direct effects. Alternatively, other experimental approaches such as rapid conditional inactivation of a gene product may be applied (Haruki *et al*, 2008; Nishimura *et al*, 2009).

Although the focus of this study has been on analyzing gene expression patterns, it is important to emphasize that other phenotypes may similarly be indirectly influenced by the associated shift in cell cycle populations. The yeast gene deletion strains studied here (Winzeler *et al*, 1999) are used in a wide variety of studies. As with the expression pattern of the entire culture, any cellular characteristic that differs dependent on the proportion of cells at specific cell cycle stages will appear apparently changed in strains with a growth rate difference. Independent of whether the readout is gene expression or a different phenotype, it is obvious that all perturbation-based studies that are accompanied by growth rate changes should take cell cycle population shifts into account. It is likely that this effect is also important for other organisms, as well as disease states associated with changes in cellular growth.

# Materials and Methods

### DNA microarray expression data

Deletion strain DNA microarray data (Fig 1) are from Kemmeren *et al* (2014), and these data are available in ArrayExpress (accession numbers E-MTAB-1383, E-MTAB-1384), GEO (accession numbers GSE42526, GSE42527) as well as in other formats described in the original study. Deletion strain data were generated as four replicates: two biological replicates harvested in early mid-log phase in synthetic complete medium with 2% glucose and each profiled in technical replicate. These were averaged and statistically compared to the average of over 400 wild-type expression profiles generated in parallel (Kemmeren *et al*, 2014). Expression data of strains grown in deletion strain conditioned medium (Fig 2) have been deposited in ArrayExpress (E-MTAB-2218) and GEO (GSE54539), as has the heat shock profile (Fig 3, E-MTAB-2219, GSE54528), the heat shock time course (Fig 4, E-MTAB-2219, GSE54528), the low phosphate response (Fig 6, E-MTAB-2217, GSE54527), and the cold shock time course (Supplementary Fig S6, E-MTAB-2219, GSE54528). Growth of strains for the experiments specific to this study is described below. For all microarray experiments, each measurement point is the average of two biological replicates, each profiled as a technical replicate in dye-swap, yielding four replicates that were averaged and statistically analyzed by Limma (Smyth, 2005) versus either wild-type or wild-type at $t = 0$. All procedures were identical and are described in detail in Kemmeren *et al* (2014), with protocols submitted to Array-Express. Calculations were made using the statistical language R version 3.0.1 on a Linux machine running CentOS 5.5, with packages and scripts provided in the Supplementary R Packages. The expression data (Miller *et al*, 2011) (Fig 3E) was obtained from ArrayExpress (accession number E-MTAB-439) and normalized with *rma* from the R package *affy*. For this data, expression changes are the $\log_2$-ratios relative to the median of the four $t = 0$ wild-types.

### RNAseq data

The Van Dijk *et al* (2011) data (Fig 3F) were obtained from Sequence Read Archive (accession number SRP005955). Reads were mapped with *bowtie* (Langmead *et al*, 2009) to the yeast genome R64.1.1 with standard settings. Read counts per gene were calculated with *featureCounts* from the R package *Rsubread* (Liao *et al*,

2014). Normalization and log ratios were obtained using the R package *edgeR* (Robinson *et al*, 2010).

### Growth rates

The doubling times of the deletion strains were calculated from the slope of the $\log_2(OD_{600})$ by taking the linear part of the growth curve just prior to harvest. For examples, see Supplementary Fig S3. The doubling times from two biological replicate cultures were averaged, and the ratio versus wild type (Supplementary Dataset S2) determined from wild-type cultures grown in parallel to each batch of mutants.

### Principal Component Analysis (PCA)

The procedure separating the slow growth profile from specific effects was performed with singular value decomposition (SVD) but is more easily referred to using the terminology from the more familiar PCA to which it is closely related. Given the full data matrix $M$, consisting of the $\log_2$-ratios (relative to wild-type) of $m$ transcripts x $n$ deletion strains we write:

$$M = U \times S \times V^T \qquad \text{(SVD formulation)}$$

or, alternatively:

$$M = X \times V^T \qquad \text{(PCA formulation)}$$

with the columns of $U$ the left-singular vectors, those of $V$ the right-singular vectors (or, in PCA terms, the principal component axes or loadings), and $S$ the diagonal matrix of singular values, $X$ the principal component scores, and $\times$ and $^T$ denoting matrix multiplication and transposition, respectively. Columns of all matrices are sorted by decreasing importance. If we rewrite, in the SVD expression, the diagonal of matrix $S$ as a simple vector $s$, it is clear that SVD decomposes matrix $M$ into a weighted sum of simpler matrices (sometimes called its modes (Alter *et al*, 2000)), as:

$$M = \sum_{k=1}^{n} s_k U_k \times V_k^T$$

with $U_k$ and $V_k$ the $k$-th column of matrices $U$ and $V$ and $k$ ranging over the number of deletion strains. Using (or conversely, discarding) a subset of dimensions allows the separation of specific features of the data. In the current study, the first mode (principal component) is identified with cell cycle phase distribution effects and is studied (Fig 1) by only using $k = 1$ or, conversely, discarded (Fig 6) by setting $s_1$ to zero. Using only $k = 1$, we obtain the first mode, $M^{(1)} = s_1 \cdot U_1 \times V_1^T$, also shown in Fig 1B. Since all $M^{(1)}$ columns are collinear, each one of them could be called the slow growth profile. For easier comparison with the actual data, we call the one with the largest norm *the* slow growth signature (Supplementary Dataset S4). The equivalent procedure in PCA terms is to create matrix $X^{(1)}$ from the scores matrix $X$ but setting all but the first column to zero, and forming:

$$M^{(1)} = X^{(1)} \times V^T$$

Conversely, the transformed data $\boldsymbol{M}^*$ is obtained by leaving out the first mode, as:

$$\boldsymbol{M}^* = \sum_{k=2}^{n} s_k U_k \times V_k^T$$

or, equivalently but more simply:

$$\boldsymbol{M}^* = \boldsymbol{M} - \boldsymbol{M}^{(1)}$$

The equivalent procedure in PCA terms is to create matrix $\boldsymbol{X}^*$ from matrix $\boldsymbol{X}$ by setting its first column to zero, and then forming $\boldsymbol{M}^* = \boldsymbol{X}^* \times \boldsymbol{V}^T$, or again using $\boldsymbol{M}^* = \boldsymbol{M} - \boldsymbol{M}^{(1)}$.

To remove the correlation with the slow growth signature from other data, each of their expression profiles was transformed in Gram-Schmidt fashion by subtracting from them their projection onto the basis vector given by the normalized slow growth profile (which is identical to the first left-singular vector $U_1$). The original (untransformed) data set, the slow growth profile, and the transformed data set are provided in Supplementary Datasets S3, S4 and S5, respectively. A single file combining the original and transformed data sets and including the *P*-values from the original measurements is also available: Supplementary Dataset S6. R scripts are provided that determine and subtract the first principal component from any data set (svd-transform.R), as well as transforming any data using the slow growth profile determined in this study (remove.signature.R). The similarity to the slow growth profile (Fig 1D) was expressed as the inner product of the strain's expression profile and the first left-singular vector $U_1$. For centered data, this is equivalent to using the covariance.

## Culture of cells

For pre-conditioned media experiments, wild-type and deletion strains were grown for two cell doublings in synthetic complete (SC) media containing 2% glucose. Cells were then harvested by centrifugation at 3,952 × *g* for 3 min. The supernatant was decanted and filtered through a 0.2-µm filter. These pre-conditioned media were then added to wild-type cells which were also allowed to grow for two cells doublings before harvesting. For the low phosphate experiment, cells were grown for two cell doublings in SC media containing 2% glucose and 75 µM phosphate before harvesting. For heat and cold shock experiments, wild-type cells were grown in SC media containing 2% glucose at either 30°C or 37°C for two cell doublings, harvested by centrifugation (3,952 × *g* for 3 min) and then re-suspended in either 37°C or 30°C media respectively depending on whether a heat or cold shock experiment was performed. Cells were then harvested at the indicated time points by centrifugation. In all cases, cells were harvested and RNA isolated and processed for microarray profiling as described previously (Kemmeren *et al*, 2014).

## Flow cytometry analysis

Cells were grown for two cell doublings and then harvested by centrifugation at the indicated time points. Cells were first fixed in 70% ethanol overnight at 4°C, then were washed twice in 1 ml flow cytometry buffer (200 mM Tris, 20 mM EDTA), re-suspended in

100 µl ribonuclease A (1 mg/ml in flow cytometry buffer; Sigma), and incubated for 2 h at 37°C in a shaking heating block at 800 rpm. Cells were washed in phosphate-buffered saline (PBS) and stained in 100 µl propidium iodide (50 µg/ml in PBS; Molecular Probes) for 1 hour at RT. Sample volume was increased to 1 ml with PBS and sonicated for 10 sec at 25% amplitude (Hielscher UP200S). DNA content was quantified by flow cytometry (FACSCalibur) and analyzed using CellQuest 5.2.

## Deconvolving the ESR using cell cycle phase-specific profiles

We consider the heat shock profile of an asynchronous population to be the sum of 'pure' cell cycle phase-specific profiles, weighted by their fraction in the population. To decompose the Gasch *et al* (2000) heat shock profile into such a weighted sum of cell cycle phase-specific profiles, we used the Spellman *et al* (1998) elutriation data as the best approximation to such 'pure' cell cycle phase-specific profiles (see also Discussion). The elutriation data comprise 14 time points from 0 till 390 min at 30-min intervals. The elutriation profiles, the experimental heat shock profile as well as the virtual heat shock profile are defined and expressed as *M*-values, that is, as the $\log_2$ of expression changes, relative to $t = 0$. Missing data were imputed with the R package *pamr* (Hastie *et al*, 2013) using default settings. We attempted to find the weights such that the weighted sum of the cell-cycle-specific time points yields a virtual expression profile most closely resembling the heat shock profile of the 859 ESR genes (Gasch *et al*, 2000). To preclude overfitting and to ensure smooth behavior at neighboring time points, a single cubic spline governing the weights was used for all 859 genes together. This spline was constructed from four control points with *x*-values chosen at 78, 156, 234, and 312 minutes. These four time points were not measured in the elutriation experiment, but provide an unbiased placement at fractions 0.2, 0.4, 0.6, and 0.8 of the duration of the elutriation data cell cycle. The *y*-values of the four control points were optimized such that the spline through them yields 14 weights (i.e., population fractions, between 0 and 1 and summing to one) that, when used in the weighted sum, give the maximal correlation of the virtual profile with the heat shock profile. (The four *y*-values themselves have little significance; *e.g.*, unlike the weights they need not be between zero and one). The deconvolution procedure was written in R and uses its built-in spline generating function *splinefun* and optimizer *optim*, both with default settings (Forsythe-Malcolm-Moler and Nelder-Mead, respectively). The optimizer is run at least ten times (with the four control points initialized randomly between 0 and 1), but more if the solutions failed to converge as judged by the best three correlations having a coefficient of variation greater than 0.01 or any of the three pairwise correlations of the top three sets of weights being below 0.95.

To rule out overfitting, the deconvolution was also performed with randomized ESR gene labels. The mean correlation of the heat shock expression data with the fit obtained after 100 randomizations of ESR gene labels was 0.06, maximum 0.13. Similarly, 100 randomizations of the cell cycle expression data matrix (each time-point column independently randomized) yield an average correlation of 0.05, maximum 0.12. The method was also cross-validated by using half the genes as a training set to obtain the four control points of the spline to weigh the time-course data. This was then applied to the other genes (test set) to determine their correlation with the heat

shock data. The mean correlation of 100 such random splits of the 859 ESR genes into training set, and test set was 0.88, that is, the same as that of the full deconvolution. The procedure is fast (seconds), general, and is made available as R package *dccd* in the Supplementary R Packages.

**Cell cycle simulation**

We modeled the distribution of cells over the phases of the cell cycle as a circular array of slots, each representing 1 minute along the cell cycle and containing the relative number of cells at that point in the cell cycle. The simulation proceeds in time steps of one minute, and cells are transferred from one slot to the next. The distribution of relative numbers of cells is initially uniform over the whole circular array (representing an unsynchronized population) and remains so if there is no G1 arrest due to heat shock. At $t = 0$, a heat shock is applied which erects a barrier at the START checkpoint in G1, between slots 14 and 15. Cells can only pass this barrier after having recovered for 25 min. The result is that the slots beyond START 'empty out', whereas the slot in front of START 'piles up' the recovering cells. For purposes of visualization, the one-minute slots were averaged into coarse-grained slot of 7 min. The lengths of the cell cycle phases, in minutes, were: G1: 21; S: 14; G2: 14; M: 21; M.G1: 7. Cells in slots 1 till 28 were deemed to have 1N DNA content, the rest 2N. For modeling the global expression changes during heat shock (Fig 4G), only fold-changes in cell numbers of greater than 1.7 were taken into account, analogous to the 1.7-fold-change threshold applied to the expression data. All parameters used in the simulation were chosen to mimic those observed experimentally, but qualitatively, the results are not sensitive to their exact values. The simulation procedure is available in the R package *s3c2* in the Supplementary R Packages.

**Supplementary information** for this article is available online: http://msb.embopress.org

### Acknowledgements

Supported by the Netherlands Bioinformatics Centre (NBIC) and the Netherlands Organization of Scientific Research (NWO), Grants 016108607, 81702015, 05071057, 91106009, 021002035 (TLL), 863.07.007 (PK), 864.11.010 (PK), 70057407 (JJB).

### Author contributions

EOD devised and performed experiments, carried out data analysis, wrote the manuscript and made figures. PL devised and carried out data analysis, wrote the manuscript and made figures. JJB, TTL, DvL, MJAGK, TM, MOB and PK contributed data. DvL and MOB performed experiments. FCPH supervised the study and wrote the manuscript.

### Conflict of interest

The authors declare that they have no conflict of interest.

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
