## [Review Process File · Molecular Systems Biology]

Cell cycle population effects in perturbation studies

Eoghan O'Duibhir, Philip Lijnzaad, Joris J. Benschop, Tineke L.Lenstra, Dik van Leenen, Marian J.A. Groot Koerkamp, Thanasis Margaritis, Mrs. Mariel O. Brok, Patrick Kemmeren, Frank C.P. Holstege

Corresponding author: Frank C.P. Holstege, UMC Utrecht

Review timeline:

Submission date:	01 February 2014
Editorial Decision:	09 March 2014
Revision received:	08 May 2014
Accepted:	12 May 2014

Editor: Thomas Lemberger

Transaction Report:

1st Editorial Decision

09 March 2014

Thank you again for submitting your work to Molecular Systems Biology. We have now heard back from the three referees who agreed to evaluate your manuscript. As you will see from the reports below, the referees find the topic of your study of potential interest and are cautiously supportive. They raise, however, several important concerns on your work, which should be convincingly addressed in a major revision of the manuscript.

One of the major points raised by reviewers #1 and #3 refers to the need to demonstrate more convincingly that changes in the cell cycle phase distribution underly the observed common 'slow growth expression signature'. Both reviewer #1 and #3 make important and constructive suggestions for additional analyses to support these claims more rigorously.

If you feel you can satisfactorily deal with these points and those listed by the referees, please submit a revised version of your manuscript. Please attach a covering letter giving details of the way in which you have handled each of the points raised by the referees. The revised manuscript might be once again subject to review.

REFeree REPORTS:

Reviewer #1:

In this work the authors analyze mRNA expression patterns of over 1000 yeast mutants, each deleted for a single gene. They identify a common slow growth expression signature, similar to the previously identified Environmental Stress Response (ESR) signature. The authors attribute this

signature to changes in the distribution of cells over different cell cycle phases in different growth rates, and in support show that it can be recapitulated to a high degree using published data of gene expression taken at different stages of the cell cycle.

The subject matter of this paper is very interesting, with broad implications for any study that involves changes in growth rate. It joins a recently revived interest in the interconnection between gene expression and growth rate, and the ways to decouple global growth-related effects from specific regulation.

This work adds both valuable data and a fresh perspective to the connection between gene expression and growth rate, by several means. First, whereas most previous works changed the growth rate by changing environmental conditions, this work explores this connection in deletion mutants. The authors find strikingly similar expression patterns when changing the growth rate by either deletion or environment, thereby increasing the generality of the phenomenon. Second, the authors add an important layer of understanding to the connection between gene expression and growth rate by suggesting that it may be largely accounted by different fractions of cells at different stages of the cell cycle in different growth rate regimes.

Altogether, the authors make a valuable contribution to the field that will be of interest to the readership of *Molecular Systems Biology*. However, I do have several major concerns regarding experimental procedures, analysis and presentation, which are detailed below.

Major points:

1. As I understand from the experimental procedures detailed in Kemmeren et al., strains were grown in liquid media to stationary (for a day), and then inoculated into fresh media. Gene expression was then profiled after two generations of exponential growth. This procedure was used for assaying both mutants and various environmental conditions. Importantly, using this experimental procedure expression is assayed when the cells are not yet in balanced growth. Balanced growth is generally assumed after 10 doublings. At the time the authors are assaying the cells a considerable fraction of the population has probably not yet recovered from stationary and did not start dividing yet. Thus, the authors cannot decouple whether the increased G1 population observed in slow-growing mutants/conditions is due to the changes in growth rate or to changes in recovery from stationary. Perhaps the 'slow-growth expression pattern' is actually a 'stationary expression pattern'? If this is the case, then the 'slow-growth expression pattern' is mostly derived from an experimental artifact and its utility for the community is limited. I think to make the authors' claim general and strong they should repeat the experiment for several strains under balanced growth conditions (in either chemostats or after 10 generations of exponential growth).
2. The authors make a highly general claim based on a biased set of mutants. The manuscript is somewhat misleading in that it states that 1484 yeast deletion strains were examined, generating the impression of a randomly sampled set. Only when reading the manuscript by Kemmeren et al., one finds that this set is focused on gene expression regulators. The authors should present the set properly, such that the readers will be aware that it is biased. Furthermore, given this bias the authors need to invest more analysis in convincing that the effect is not dominated by the regulatory nature of their dataset. Such an attempt has been made in figure 1c, however I do not find this analysis convincing. On the contrary, figure 1c shows that there are substantial differences in the representation of some of the categories in the group showing the slow-growth expression pattern (for example protein trafficking, which appears over-represented, and gene-specific transcription factor, which appears under-represented). Whereas I am convinced by the correlation between growth rate and the slow-growth expression pattern, the GO analysis presented is not convincing that the effect is not dominated by specific groups. Incidentally, both results can coincide if, for example, deletions of genes with similar functions result in both similar growth rate and similar expression patterns. If the authors want to convince that *gr* plays a greater role than GO they should perform additional analyses, for example, show that correlations between pairs of deletions with similar growth rates, but belong to different GO categories are generally higher than correlations between pairs from the same GO category, but that result in different growth rates. Also, they can select from their set random subsets that recapitulate the genomic distribution of GO categories, and examine whether their results still hold to control for the initial biases in the examined set. I would also add some supplementary figures and analyses to examine other factors, which may be attributed to the common effect, such as average expression level, connectivity of the protein in protein-

interaction networks etc., and examine whether any of these have a better explanatory power than growth rate. If these have some explanatory power, then it is worth examination, and if all have less explanatory power than growth rate it will make the authors' claim much more convincing.

3. The authors fit weights to 14 cell cycle phases to obtain maximum correlation to a particular expression pattern (Heat shock, 15min) and then state that the high correlation observed indicates that expression patterns are largely determined by cell cycle population shifts. The analysis performed is likely overfitting as many parameters are being optimized (4 cubic splines). Unfortunately, the resulting parameters are not subjected to further quantitative testing or cross validation. The agreement with the flow cytometry data is only qualitative (more/less cells in G1) with no numerical indication of the proportions. Numbers of fractions of cells in G1/S/G2+M should be indicated for both model and flow cytometry measurements and compared, for both heat-shock and mutant experiments. Quantitative agreement will reinforce the authors' claim, whereas disagreement will indicate that the initial high correlation was indeed a result of overfitting. These numbers should also be discussed in light of previous literature that looked at fraction of cells in different stages of the cell cycle in different growth conditions/mutants. High deviations from previously-described fractions of cells in different stages of the cell cycle may indicate that indeed the experimental setup used in this work (in which the cells are not in balanced growth - see comment 1), increases the 1N population and therefore the impact of the reported expression signature.

4. The authors compare their results extensively to the previously defined ESR. However, whereas the ESR was defined more than a decade ago there has been a body of work since that attributed much of the ESR to changes in growth rate, as also acknowledged by the authors in the discussion. There has also been much work in E.coli that connected growth rate to many cellular parameters, including gene expression. As such, to make the work more relevant to current knowledge, the authors should focus less on the ESR and discuss whether their slow growth signature in mutants is similar to the slow growth signature observed in WT strains in different growth conditions.

5. Figure 1d- the authors claim that the points with lower correlation (off-diagonal) are due to additional gene expression changes specific to those individual mutants. This statement is not backed by any analysis. The authors should present the names of these mutants, provide examples for these 'specific' expression changes and explain why they are interpreted as specific. It should be explained what is common to these deletions. Why do they exhibit more changes over the prevailing growth-rate signature compared with other deletions? Do they belong to a specific GO category? Is this significant? Are they relatively upstream in signaling networks? Are they more connected in protein-protein interaction networks?

6. Presentation of experimental procedures and figure legends are severely lacking. Even if complete procedures were previously described in other papers, the manuscript should include a short recapitulation of the main experiments and analysis performed. Similarly for figure legends. The appropriate sections should be augmented.

Minor points:

1. Figure 1- Legend is lacking. Many details that appear in the figures are not specified in the legend. For example, an explanation regarding color code for the points in figure 1d is missing (what are the blue dots and what are the gray?)
2. Figure 1c- p-values should be added to the analysis and properly presented in either text or figure.
3. The introduction does not clearly state the goal of this work.
4. The analysis of medium depletion is a valuable control, however the results are neither surprising nor extremely interesting. I would consider moving this section to the supplementary to allow room for the more important analyses.
5. FACS is an acronym for Fluorescence Activated Cell Sorting. The authors have not performed sorting in this work and therefore should use the appropriate term- flow cytometry measurements.

Reviewer #2:

In this manuscript O'Duibhir et al. present an elegant method to identify and correct for the effects of cell-cycle variations in gene expression data. The study convincingly proves that the transcriptional effect observed in many stress conditions and yeast deletion mutants can be

explained simply by the redistribution in number of cells at different cell cycle stages associated to a slow growth phenotype. The method described here will be ubiquitously applicable to any data set analyzing gene expression across different genotypes or phenotypes and for other organisms as well. And it will be especially useful to disentangle direct effects from downstream consequences due to changes in cellular growth. Since I see that this method could be widely used, I would recommend the acceptance of this manuscript after the authors address a few key points in the discussion that will further enrich the manuscript.

- 1) Firstly, what is relationship between the signature of the cell-cycle vector with the platform used to measure gene expression. It is clear from the paper that when applying the method to datasets such as Gasch et al. and Kemmeren et al., using different array technology, the results vary a bit. A brief discussion on how a change in platform might affect the results and may be accounted for should be discussed.
- 2) Along the same line, in order to prove the ubiquity and platform independence of the method, it would be desirable that the authors demonstrate that their method is also applicable to previously published RNA-Seq data. As that is the most common technology used nowadays.
- 3) Although the authors mention ESR genes to be a part of the cell-cycle signature vector, an expanded discussion about which genes are enriched in the cell-cycle signature, GO terms analysis would shed light on why the slow growth phenotype might manifest as a result of stress and in different genotypes.
- 4) As a minor note, I am not sure if the authors used 2 μ m (or rather 0.2 μ m) filters to obtain the pre-conditioned media.

Reviewer #3:

Applying principal component analysis on a large dataset of yeast mutant transcription profiles, O'Duibhir and colleagues report a general transcriptional response to gene deletion that is similar to the early stress response seen upon environmental perturbation. This main response also correlates with growth rate and with distribution of cells along the cell cycle phase (proportion of 1N versus 2N-cells from FACS data). It is proposed that this general transcriptional effect at the population level is the consequence of a change in the distribution of cells among the cell cycle phases. This hypothesis is corroborated by a computational deconvolution of the steady-state mutant expression profiles over phase-specific gene expression data from a cell-cycle study. Finally, a method to remove this main effect from transcription profiles is provided and it is demonstrated that it helps distinguishing the direct effect of genetic perturbations from indirect effects likely due to change in growth behavior.

The finding that the transcriptional effect of genetic perturbation resembles the effect of environmental perturbation is not very surprising (Fig. 1-3). Also, the correlation between transcriptional response to stress and growth rate had already been reported (as properly acknowledged in the manuscript: Brauer et al for instance). Nevertheless, the proposed method to remove this effect is useful for the yeast community and beyond (Fig. 6). These claims are well supported and deserve publication. However, the most innovative aspect is the change in cell phase distribution (Fig. 4). It is provocative because it suggests that there is no general transcriptional stress response by itself that is not explained by a change in the distribution of cells among the phases of the cell cycle. In response to stress, cells would temporarily arrest in the G1 phase, thereby inducing a change in phase distribution within the population and thus an overall change of expression at the population level. It is very surprising because there are well documented general stress response genes (TATA-containing genes, stress-activated protein kinase (SAPK) and TOR pathway, see also the excellent review LÚpez-Maury et al. *Nat. Genet.* 2008), which are thought to be distinct from G1 phase genes. However, this claim is not very well supported. Although the paper could be accepted without it, the authors should try to make this analysis more convincing.

Indeed this analysis (Fig. 4) suffers from the following drawbacks:

- 1) The "G1" phase expression data likely contains a superposition of non-cycling stress response signal and of unstressed cycling G1 signal. Indeed, the phase-specific expression levels were taken

from one cell-cycle time-series (Elutriation series, Spellman et al. 1998). This series, as other cell-cycle time-series, is based on a synchronization protocol (in this case based on centrifugation), which could stress the cells at the initial time point. Hence, the first time points of synchronized populations often present a transcriptional response that is not cyclic (Spellman et al. 1998, Guo et al PNAS 2013). Moreover, the elutriation series from Spellmann et al. starts in the G1 phase. Thus, the G1 phase is the one most likely containing non-cyclic stress response signal. The G1 phase data at the next cycles could not be considered for the elutriation series because it covered a single cell cycle only.

2) The algorithm inferring the proportion of cells in each phase is not formerly described (detailed concerns below). A formal description and an implementation should be provided.

To overcome these issues, a few approaches could be investigated:

1) Using other cell cycle data. Spellmann et al published two other time series that cover at least two cell cycles. Granovskaia et al, Genome Biol. 2010, reported two further datasets with at least two cell cycles with amore recent technology (high resolution tiling array). Data from the later cycles, which are less prone to have overlapping stress response signal could be used.

2) Using the fit of Guo et al. PNAS 2013. These authors have developed a computational method to extract the pure expression levels of each cell cycle stage from cell-cycle time series. They removed non-periodic signal found in the early stages, controlled for asynchrony and distinguished daughter from mother cells.

My personal conviction is that the first component identified here is a mix of general stress response and of response to change in growth. The distinction between these two components has been so far elusive (López-Maury et al. Nat. Genet. 2008). By breaking down the first component into a cell-cycle phase part and an orthogonal "stressed" part (expected to be TATA-rich, in TOR pathways, etc.), this study could be able to dissect and quantify the respective contribution of each to the global stress response.

Detailed comments

p.2 These two statements are unclear: "The challenge with regard to indirect effects is that typically these are not of a general nature." "Depending on the goals of a particular study, indirect effects nevertheless need to be taken into account, especially if the goal is to derive molecular mechanisms."

p.3 SVD and principal component analysis are formally the same (the eigenvectors of the right/left space are those of the covariance matrix / transpose of the matrix). SVD is a mathematical decomposition, PCA is a statistical method based on it. Hence, I suggest using the PCA terminology (what is done is the analysis of the first principal component). Also, readers will be more familiar with PCA.

Fig. 1C "chromatin factor" => chromatin factor, "transcription machinery" => transcription machinery. Plotting the odd ratios or sorting the categories by decreasing ratio $\#\{r>0.5\} / \#\{\text{all}\}$ will better highlight the important categories.

Fig. 2F-I. The reproducibility is remarkable. The authors should describe all pre-processing of the raw data including the normalization.

Fig. 3A could be replaced by the projection of each of these datasets to the plane of the two first principal components (PC1, PC2). We shall see Gasch and Causton far on the right (along PC1) but also be able to visualize all other datasets and maybe understand what the second component is.

Fig 3B: "Correlation of the slow growth..." It is not a correlation but a scatter plot. Applies to 4A and 5A as well.

Fig. 4: Also here a panel could show the projection of each time point of Spellmann et al on (PC1,PC2).

Fig. 4A should show all data points, as the model was fitted on all data.

Methods "modeling of the ESR cell cycle phase signatures":

The statistical model should be formally described. In particular it is unclear:

- i) what was fitted. Was it the natural (unlogged) gene expression values (concentrations should sum up in the natural scale) or something else?
 - ii) how the reference signal (wild type unsynchronized population) was taken into account
 - iii) what the noise model was. Indeed variance of unlogged microarray gene expression data grows with expression value (heteroskedasticity) so if sums on the natural scale were modeled (i), this point must be addressed.
 - iv) whether the proportions were constrained to be non-negative and sum to 1
 - v) what exact 4 time points were taken as spline nodes
 - vi) which optimization algorithm was used.
- Moreover, the corresponding scripts should be provided.

1st Revision - authors' response

08 May 2014

All three referees agree on the general importance of our findings. Several issues were raised which we have extensively addressed. Our answers to the specific points are in italics below. In red italics we state the changes made to manuscript. We thank the referees for all their suggestions. The additional analyses and the changes made have considerably improved the study.

Reviewer #1:

In this work the authors analyze mRNA expression patterns of over 1000 yeast mutants, each deleted for a single gene. They identify a common slow growth expression signature, similar to the previously identified Environmental Stress Response (ESR) signature. The authors attribute this signature to changes in the distribution of cells over different cell cycle phases in different growth rates, and in support show that it can be recapitulated to a high degree using published data of gene expression taken at different stages of the cell cycle.

The subject matter of this paper is very interesting, with broad implications for any study that involves changes in growth rate. It joins a recently revived interest in the interconnection between gene expression and growth rate, and the ways to decouple global growth-related effects from specific regulation.

This work adds both valuable data and a fresh perspective to the connection between gene expression and growth rate, by several means. First, whereas most previous works changed the growth rate by changing environmental conditions, this work explores this connection in deletion mutants. The authors find strikingly similar expression patterns when changing the growth rate by either deletion or environment, thereby increasing the generality of the phenomenon. Second, the authors add an important layer of understanding to the connection between gene expression and growth rate by suggesting that it may be largely accounted by different fractions of cells at different stages of the cell cycle in different growth rate regimes.

Altogether, the authors make a valuable contribution to the field that will be of interest to the readership of Molecular Systems Biology. However, I do have several major concerns regarding experimental procedures, analysis and presentation, which are detailed below.

Major points:

1. As I understand from the experimental procedures detailed in Kemmeren et al., strains were grown in liquid media to stationary (for a day), and then inoculated into fresh media. Gene expression was then profiled after two generations of exponential growth. This procedure was used for assaying both mutants and various environmental conditions. Importantly, using this experimental procedure expression is assayed when the cells are not yet in balanced growth. Balanced growth is generally assumed after 10 doublings. At the time the authors are assaying the cells a considerable fraction of the population has probably not yet recovered from stationary and did not start dividing yet. Thus, the authors cannot decouple whether the increased G1 population observed in slow-growing mutants/conditions is due to the changes in growth rate or to changes in recovery from stationary. Perhaps the 'slow-growth expression pattern' is actually a 'stationary

expression pattern'? If this is the case, then the 'slow-growth expression pattern' is mostly derived from an experimental artifact and its utility for the community is limited. I think to make the authors' claim general and strong they should repeat the experiment for several strains under balanced growth conditions (in either chemostats or after 10 generations of exponential growth).

The referee is concerned that the mutants with slower growth may have a different flow cytometry profile due to insufficient recovery time from the overnight culture, rather than due to a slower cell cycle in general. This is more applicable to bacterial culture than to yeast. Overnight pre-culture of bacteria typically result in stationary phase. Overnight pre-culture of yeast typically go into diauxic shift, rather than real stationary phase which takes 5-9 days to achieve for yeast (see PMID 15837421). The recovery time from diauxic shift is fast. When setting up the procedures for analyzing the mutants we nevertheless considered the concern raised by the referee and found that two cell doublings after overnight pre-culture is sufficient to achieve balanced, exponential growth, also for slow growing mutants. Shown below are the growth curves prior to harvesting for the slowest growing strains from Figure 5. The blue dots are the number of doublings. Orange dots are the doubling times of the culture (inverse of slope of blue dots) as determined with a sliding window of ten consecutive culture measurements. By the time of harvesting (last blue point, ± 1 hour after the last orange point), all mutants are out of lag-phase and have achieved different rates of balanced growth. The concern that slow growing mutants may not have attained balanced growth at the time of harvest is therefore ruled out.

Since other scientists may have the same concern we now mention these observations in the Results (page 4, paragraph 2) and have added this figure as supplementary figure 3, also displayed here:

2. The authors make a highly general claim based on a biased set of mutants. The manuscript is somewhat misleading in that it states that 1484 yeast deletion strains were examined, generating the impression of a randomly sampled set. Only when reading the manuscript by Kemmeren et al., one finds that this set is focused on gene expression regulators. The authors should present the set properly, such that the readers will be aware that it is biased. Furthermore, given this bias the authors need to invest more analysis in convincing that the effect is not dominated by the regulatory nature of their dataset. Such an attempt has been made in figure 1c, however I do not find this analysis convincing. On the contrary, figure 1c shows that there are substantial differences in the representation of some of the categories in the group showing the slow-growth expression pattern (for example protein trafficking, which appears over-represented, and gene-specific transcription factor, which appears under-represented). Whereas I am convinced by the correlation between growth rate and the slow-growth expression pattern, the GO analysis presented is not convincing that the effect is not dominated by specific groups. Incidentally, both results can coincide if, for example, deletions of genes with similar functions result in both similar growth rate and similar

expression patterns. If the authors want to convince that gr plays a greater role than GO they should perform additional analyses, for example, show that correlations between pairs of deletions with similar growth rates, but belong to different GO categories are generally higher than correlations between pairs from the same GO category, but that result in different growth rates. Also, they can select from their set random subsets that recapitulate the genomic distribution of GO categories, and examine whether their results still hold to control for the initial biases in the examined set. I would also add some supplementary figures and analyses to examine other factors, which may be attributed to the common effect, such as average expression level, connectivity of the protein in protein-interaction networks etc., and examine whether any of these have a better explanatory power than growth rate. If these have some explanatory power, then it is worth examination, and if all have less explanatory power than growth rate it will make the authors' claim much more convincing.

There is a misunderstanding about what we meant to achieve with this analysis (figure 1C) and how the referee has interpreted our intention. Clearly we need to rephrase this section.

First, it was not our intention to present the 1484 deletion strains as a randomly chosen set. In fact, figure 1C already showed which different functional categories are represented.

To make it clearer that this is not a random set, we now also describe the set in the Results (page 3, paragraph 2).

Second, we are not trying to conclude that GO plays no role (two categories are over-represented). The only conclusion that we draw from figure 1C is that the common gene expression signature is not restricted to one particular functional category. Rather, mutants with this signature represent all kinds of different functional classes. The reason for drawing attention to this is because in the past similarity between expression signatures of mutants has been interpreted as indicating shared function. There are still many other signatures that indicate shared function (>75% of the changes in expression, our answer to point 5 below exemplifies this further) but the common signature described here is not one of these because it can be found upon deletion of strains from many different functional categories (figure 1C).

We have rewritten the section, also incorporating the suggestions made under point 5 below to include examples of the off-diagonal mutants with specific effects (page 3, paragraph 3 – page 4, paragraph 1).

Although the mutants analyzed are focused on regulators, the scope is still broad (one quarter of all yeast genes) and covers many different functional groups (figure 1C). Members of any group can exhibit the common signature (figure 1C). The focus on regulators (eg transcription factors, chromatin regulators) would actually bias the signatures to encompass more direct effects rather than the indirect one related to slow growth. Since the same effect is also observed in wild type cells subjected to different (slow) growth conditions, the nature of the mutants analyzed here has not influenced the general conclusion that slow growth results in an apparent expression response that is actually the result of a cell cycle population shift.

3. The authors fit weights to 14 cell cycle phases to obtain maximum correlation to a particular expression pattern (Heat shock, 15min) and then state that the high correlation observed indicates that expression patterns are largely determined by cell cycle population shifts. The analysis performed is likely overfitting as many parameters are being optimized (4 cubic splines).

The original description of the deconvolution procedure was too concise. Overfitting is unlikely. Only 4 parameters were varied to obtain a single fit for the expression levels of 859 ESR genes. These 4 parameters are the y-values of the four control points of the single spline that governs the weighting of the 14 different time points.

A more extensive description of the procedure has now been provided in the Materials and Methods (page 16, paragraph 1). The description in the Results has also been extended to avoid this confusion (page 6, paragraph 1). The scripts are also made available.

Unfortunately, the resulting parameters are not subjected to further quantitative testing or cross validation.

We have rectified this by applying the same procedure to 100 different randomizations of the gene labels of the ESR genes in the heat-shock expression profile. This yields an average correlation of

0.06 (maximum 0.13). Compared to the original correlation of 0.88 for the non-randomized set, this further demonstrates that the result obtained was not due to overfitting. Similarly, 100 randomizations of the cell cycle expression data matrix (each time-point column independently randomized) yields an average correlation of 0.05, maximum 0.12, again strongly arguing against overfitting. In addition to the randomizations we have also performed a cross-validation as suggested, i.e. performing the same fitting procedure with only half of the ESR genes and predicting the expression of the other half. This was repeated 100 times with random training and test sets. The average correlation for the predicted gene expression was 0.88 with a standard deviation of 0.09. Again this confirms that there is no overfitting.

These different controls are now described in the Results (page 6, paragraph 2) and Materials and Methods (page 17, paragraph 1). These procedures are also made available within the source code.

The agreement with the flow cytometry data is only qualitative (more/less cells in G1) with no numerical indication of the proportions. Numbers of fractions of cells in G1/S/G2+M should be indicated for both model and flow cytometry measurements and compared, for both heat-shock and mutant experiments. Quantitative agreement will reinforce the authors' claim, whereas disagreement will indicate that the initial high correlation was indeed a result of overfitting.

The numerical comparison suggested here was in fact already included in figure 4E (dots: experimentally derived fractions; line: fractions derived from simulation). The results also agree quantitatively. Figure 5A also already included a numerical comparison for the mutants. We have rephrased the sentences describing these results so as to draw more attention to the numerical comparisons that were already present (page 7, paragraph 2 and page 8, paragraph 2)).

These numbers should also be discussed in light of previous literature that looked at fractions of cells in different stages of the cell cycle in different growth conditions/mutants. High deviations from previously-described fractions of cells in different stages of the cell cycle may indicate that indeed the experimental setup used in this work (in which the cells are not in balanced growth - see comment 1), increases the 1N population and therefore the impact of the reported expression signature.

The cultures that we report on are in balanced growth (see answer to comment 1 above). Furthermore, our reported cell cycle population shifts during heat shock (Figure 4D) quantitatively agree with previously reported population shifts in Rowley et al., and Johnston and Singer. These papers were already cited. We have now changed the text relating to these references to clearly state this correspondence (page 7, paragraph 2 and page 8, paragraph 2). With regard to specific mutants, a complete analysis of all flow cytometry profiles for the entire gene deletion collection has (to the best of our knowledge) not yet been reported. The Lu et al., study (already cited) contained flow cytometry profiles for several deletion strains. Although a quantitative analysis of their data was not included in that study, the flow cytometry profiles that we report (Figure 5A, B) agree qualitatively with theirs. Since different strains were analyzed it doesn't make sense to refer to this.

4. The authors compare their results extensively to the previously defined ESR. However, whereas the ESR was defined more than a decade ago there has been a body of work since that attributed much of the ESR to changes in growth rate, as also acknowledged by the authors in the discussion. There has also been much work in E.coli that connected growth rate to many cellular parameters, including gene expression. As such, to make the work more relevant to current knowledge, the authors should focus less on the ESR and discuss whether their slow growth signature in mutants is similar to the slow growth signature observed in WT strains in different growth conditions.

The discussion starts with this point and we now further strengthen the connection (page 10, paragraph 1). In addition, also in response to referee 3, we have included two new panels in Figure 3 (E and F) showing the correlation for more recently generated data under different growth conditions and using different platforms.

5. Figure 1d- the authors claim that the points with lower correlation (off-diagonal) are due to additional gene expression changes specific to those individual mutants. This statement is not backed by any analysis. The authors should present the names of these mutants, provide examples

for these 'specific' expression changes and explain why they are interpreted as specific. It should be explained what is common to these deletions. Why do they exhibit more changes over the prevailing growth-rate signature compared with other deletions? Do they belong to a specific GO category? Is this significant? Are they relatively upstream in signaling networks? Are they more connected in protein-protein interaction networks?

As is pointed out in answer to comment 2 above, the slow growth associated signature accounts for only part (24%) of the gene expression changes observed in the dataset. Explaining or describing all the other mutant signatures is beyond the scope of the current manuscript, which is focused on explaining the most commonly found signature. The cited references (Hughes et al. 2000, Roberts et al. 2000, Benschop et al. 2010, Lenstra et al. 2011), as well as the paper describing the initial dataset (Kemmeren et al. 2014), contain numerous specific examples as well as general analyses of how specific expression signatures relate to specific pathways and/or protein complexes.

Given this comment we think that it may nevertheless be important to include several examples of the off-diagonal mutants as suggested. These have been included as Supplementary figure 2. These exemplify that mutants with the slow growth signature also have additional expression signatures specific to particular protein complexes and pathways, a point that is already well made in the cited literature.

6. Presentation of experimental procedures and figure legends are severely lacking. Even if complete procedures were previously described in other papers, the manuscript should include a short recapitulation of the main experiments and analysis performed. Similarly for figure legends. The appropriate sections should be augmented.

The descriptions of procedures (Materials and Methods) and legends have been extended.

Minor points:

1. Figure 1- Legend is lacking. Many details that appear in the figures are not specified in the legend. For example, an explanation regarding color code for the points in figure 1d is missing (what are the blue dots and what are the gray?)

Blue dots show the deletion mutants further analyzed by flow cytometry in figure 5. This and other missing details have now been supplied.

2. Figure 1c- p-values should be added to the analysis and properly presented in either text or figure.

This has been done (figure legend).

3. The introduction does not clearly state the goal of this work.

A sentence that concisely describes the goal is now included (page 2, paragraph 2).

4. The analysis of medium depletion is a valuable control, however the results are neither surprising nor extremely interesting. I would consider moving this section to the supplementary to allow room for the more important analyses.

Given the comment of referee#3 it has been kept an integral part of the Results.

5. FACS is an acronym for Fluorescence Activated Cell Sorting. The authors have not performed sorting in this work and therefore should use the appropriate term- flow cytometry measurements.

Corrected throughout.

Reviewer #2:

In this manuscript O'Duibhir et al. present an elegant method to identify and correct for the effects of cell-cycle variations in gene expression data. The study convincingly proves that the transcriptional effect observed in many stress conditions and yeast deletion mutants can be explained simply by the redistribution in number of cells at different cell cycle stages associated to a

slow growth phenotype. The method described here will be ubiquitously applicable to any data set analyzing gene expression across different genotypes or phenotypes and for other organisms as well. And it will be especially useful to disentangle direct effects from downstream consequences due to changes in cellular growth. Since I see that this method could be widely used, I would recommend the acceptance of this manuscript after the authors address a few key points in the discussion that will further enrich the manuscript.

1) Firstly, what is relationship between the signature of the cell-cycle vector with the platform used to measure gene expression. It is clear from the paper that when applying the method to datasets such as Gasch et al. and Kemmeren et al., using different array technology, the results vary a bit. A brief discussion on how a change in platform might affect the results and may be accounted for should be discussed.

This discussion has now been added (page 11, paragraph 3).

2) Along the same line, in order to prove the ubiquity and platform independence of the method, it would be desirable that the authors demonstrate that their method is also applicable to previously published RNA-Seq data. As that is the most common technology used nowadays.

In general the method is platform independent since results from all platforms are transformed into relative gene expression changes. Two different platforms were included originally. As a further demonstration of this we now include Figure 3E and F (referred to in the manuscript on page 5, paragraph 1): different growth conditions and technology platforms (Affymetrix and RNAseq).

3) Although the authors mention ESR genes to be a part of the cell-cycle signature vector, an expanded discussion about which genes are enriched in the cell-cycle signature, GO terms analysis would shed light on why the slow growth phenotype might manifest as a result of stress and in different genotypes.

Enriched GO categories have now been included as Supplementary Table 1 and a discussion of this has been added (page 11, paragraph 2).

4) As a minor note, I am not sure if the authors used 2 μ m (or rather 0.2 μ m) filters to obtain the pre-conditioned media.

Corrected (0.2 μ m) (page 15, paragraph 2).

Reviewer #3:

Applying principal component analysis on a large dataset of yeast mutant transcription profiles, O'Duibhir and colleagues report a general transcriptional response to gene deletion that is similar to the early stress response seen upon environmental perturbation. This main response also correlates with growth rate and with distribution of cells along the cell cycle phase (proportion of 1N versus 2N-cells from FACS data). It is proposed that this general transcriptional effect at the population level is the consequence of a change in the distribution of cells among the cell cycle phases. This hypothesis is corroborated by a computational deconvolution of the steady-state mutant expression profiles over phase-specific gene expression data from a cell-cycle study. Finally, a method to remove this main effect from transcription profiles is provided and it is demonstrated that it helps distinguishing the direct effect of genetic perturbations from indirect effects likely due to change in growth behavior.

The finding that the transcriptional effect of genetic perturbation resembles the effect of environmental perturbation is not very surprising (Fig. 1-3). Also, the correlation between transcriptional response to stress and growth rate had already been reported (as properly acknowledged in the manuscript: Brauer et al for instance). Nevertheless, the proposed method to remove this effect is useful for the yeast community and beyond (Fig. 6). These claims are well supported and deserve publication. However, the most innovative aspect is the change in cell phase distribution (Fig. 4). It is provocative because it suggests that there is no general transcriptional stress response by itself that is not explained by a change in the distribution of cells among the

phases of the cell cycle. In response to stress, cells would temporarily arrest in the G1 phase, thereby inducing a change in phase distribution within the population and thus an overall change of expression at the population level. It is very surprising because there are well documented general stress response genes (TATA-containing genes, stress-activated protein kinase (SAPK) and TOR pathway, see also the excellent review López-Maury et al. *Nat. Genet.* 2008), which are thought to be distinct from G1 phase genes. However, this claim is not very well supported. Although the paper could be accepted without it, the authors should try to make this analysis more convincing.

Indeed this analysis (Fig. 4) suffers from the following drawbacks:

1) The "G1" phase expression data likely contains a superposition of non-cycling stress response signal and of unstressed cycling G1 signal. Indeed, the phase-specific expression levels were taken from one cell-cycle time-series (Elutriation series, Spellman et al. 1998). This series, as other cell-cycle time-series, is based on a synchronization protocol (in this case based on centrifugation), which could stress the cells at the initial time point. Hence, the first time points of synchronized populations often present a transcriptional response that is not cyclic (Spellman et al. 1998, Guo et al. *PNAS* 2013). Moreover, the elutriation series from Spellmann et al. starts in the G1 phase. Thus, the G1 phase is the one most likely containing non-cyclic stress response signal. The G1 phase data at the next cycles could not be considered for the elutriation series because it covered a single cell cycle only.

Compared to the other cell cycle synchronization methods (heat-shock for *cdc28-ts*, heat-shock for *cdc15-2* and alpha-factor arrest), the elutriation dataset is likely the least influenced by stress. This advantage, as well as better inherent synchronization compared to the other methods, are extensively discussed in the paper by Shedden and Cooper (PMID 12087178). This is why we choose the elutriation-based time course data to model the ESR/slow growth effects (see further comments below).

2) The algorithm inferring the proportion of cells in each phase is not formerly described (detailed concerns below). A formal description and an implementation should be provided.

We apologize for this. Extensive description in the Materials and Methods and R packages have been added.

To overcome these issues, a few approaches could be investigated:

1) Using other cell cycle data. Spellmann et al published two other time series that cover at least two cell cycles. Granovskaia et al, *Genome Biol.* 2010, reported two further datasets with at least two cell cycles with amore recent technology (high resolution tiling array). Data from the later cycles, which are less prone to have overlapping stress response signal could be used.

As discussed by Shedden and Cooper (PMID 12087178), the elutriation method is the least prone to stress and all methods suffer from quite rapid loss of synchronization. When other cell cycle time series are used, the results are nevertheless similar. For elutriation the correlation between heat shock and the cell cycle based model (Figure 4A) was 0.88. For alpha factor arrest (Spellman/Granovskaia) the correlation is 0.82/0.63 and for *cdc28-ts* (Spellman/Granovskaia) the correlation is 0.71/0.35. We have no explanation for why the Granovskaia *cdc28-ts* yields a lower correlation compared to the same method from the Spellman dataset.

We report the correlations using other cell cycle time course data in the results (page 6, paragraph 2 – page 7, paragraph 1). We also refer to the Shedden and Cooper paper to rationalize our choice for focusing on elutriation, at the same time pointing out that none of the methods used is perfect (page 6, paragraph 2).

2) Using the fit of Guo et al. *PNAS* 2013. These authors have developed a computational method to extract the pure expression levels of each cell cycle stage from cell-cycle time series. They removed non-periodic signal found in the early stages, controlled for asynchrony and distinguished daughter from mother cells.

We attempted both this and 1) above. We gave up trying to use the Guo et al dataset mainly because option 1) worked well and also in part due to ambiguities in the description of the Guo deconvoluted data which was also in a non-standard format.

My personal conviction is that the first component identified here is a mix of general stress response and of response to change in growth. The distinction between these two components has been so far elusive (López-Maury et al. Nat. Genet. 2008). By breaking down the first component into a cell-cycle phase part and an orthogonal "stressed" part (expected to be TATA-rich, in TOR pathways, etc.), this study could be able to dissect and quantify the respective contribution of each to the global stress response.

We agree that there are likely two components and in retrospect it is clear that we were too one-sided in the discussion of our findings which inadvertently conveyed the impression that the entire ESR can be accounted for solely by the cell cycle population effect. With the cell cycle data we can account for at most 88% of the ESR based on the population shift (Figure 4A). It is indeed well known that transcription factors such as Msn2/4 are involved in a direct transcriptional response to general stresses. This may correspond to that part of the ESR that we cannot model. It is challenging to completely disentangle the two. Starting with Msn2 targets we have tried to make the distinction, but even for such a well-established factor and its direct targets this is not trivial. We think that this is due to the intimate coupling of the metabolic/redox cycle with the cell cycle: our preliminary analyses suggest that some (but not all) Msn2 targets are also cell-cycle regulated likely due to redox fluctuations during the cell cycle. A similar phenomenon also appears to occur for other stress transcription factors associated with the response to oxidative stress (e.g. Skn7). At this stage we think that it would be best to focus the paper on the conclusion that a cell cycle population shift contributes to many of the gene expression changes observed upon genetic or environmental perturbations that were previously ascribed to "growth".

We have expanded the discussion to include a section about the contribution of both cell cycle population effects (which are clearly taking place upon both environmental and genetic perturbations) and the general stress response mediated by factors such as Msn2/4. We state that more work is required to completely unravel the two, also alluding to the complications of the redox cycling described above (page 10, paragraph 3 – page 11, paragraph 1). Changes have been made throughout the results to indicate that a large part, rather than all of the ESR is due to cell cycle population effects.

Detailed comments

p.2 These two statements are unclear: "The challenge with regard to indirect effects is that typically these are not of a general nature." "Depending on the goals of a particular study, indirect effects nevertheless need to be taken into account, especially if the goal is to derive molecular mechanisms."

The first statement has been removed and the second has been changed to clarify what was meant (page 2, paragraph 1).

p.3 SVD and principal component analysis are formally the same (the eigenvectors of the right/left space are those of the covariance matrix / transpose of the matrix). SVD is a mathematical decomposition, PCA is a statistical method based on it. Hence, I suggest using the PCA terminology (what is done is the analysis of the first principal component). Also, readers will be more familiar with PCA.

We agree and have changed the terminology throughout to PCA, but keeping it clear (in the Materials and Methods) that we have applied SVD.

Fig. 1C "chromatin factor" => chromatin factor, "transcription machinery" => transcription machinery. Plotting the odd ratios or sorting the categories by decreasing ratio $\#\{r>0.5\} / \#\{all\}$ will better highlight the important categories.

Typos corrected in figure 1. We now also include p-values.

Fig. 2F-I. The reproducibility is remarkable. The authors should describe all pre-processing of the raw data including the normalization.

The reproducibility is due to automation and many years of using external control calibration standards for optimization of accuracy and precision.

More description of all growth and microarray procedures has been added (page 13, paragraph 1). Full details are included in the paper describing the primary dataset (Kemmeren et al.) that extensively describes all steps.

Fig. 3A could be replaced by the projection of each of these datasets to the plane of the two first principal components (PC1, PC2). We shall see Gasch and Causton far on the right (along PC1) but also be able to visualize all other datasets and maybe understand what the second component is.

We have done this (first figure below). We think that the original Figure 3A is clearer and that it would be confusing to introduce the second principal component at this stage since we have not investigated it in any detail. The second figure below is also of interest in this respect. Whereas PC1 captures 24% of the variation in the Kemmeren et al. data, PC2 doesn't distinguish itself from the remaining components in this respect.

Projection of compendium datasets to the plane of the two first principal components (PC1, PC2):

Supplemental Figure 1:

We think that the second figure above is a useful additional analysis from which to judge the importance and singularity of PC1. We now refer to it in the Results (page 3, paragraph 2) as Supplemental Figure 1

Fig 3B: "Correlation of the slow growth..." It is not a correlation but a scatter plot. Applies to 4A and 5A as well.

This has been rectified.

Fig. 4: Also here a panel could show the projection of each time point of Spellmann et al on (PC1,PC2).

See answer to previous comment about PC2

Fig. 4A should show all data points, as the model was fitted on all data.

The original description of what the model was fitted to (only the ESR genes) was not clear enough. This has been rectified by an extended description in the Materials and Methods:

Materials and Methods "modeling of the ESR cell cycle phase signatures":

The statistical model should be formally described. In particular it is unclear:

- i) what was fitted. Was it the natural (unlogged) gene expression values (concentrations should sum up in the natural scale) or something else?*
- ii) how the reference signal (wild type unsynchronized population) was taken into account*
- iii) what the noise model was. Indeed variance of unlogged microarray gene expression data grows with expression value (heteroskedascisty) so if sums on the natural scale were modeled (i), this point must be addressed.*
- iv) whether the proportions were constrained to be non-negative and sum to 1*
- v) what exact 4 time points were taken as spline nodes*
- vi) which optimization algorithm was used.*

Moreover, the corresponding scripts should be provided.

These points have all been incorporated in the more extensive descriptions and by making R-packages available for each method.